# The Impact of Innovative Plant Sources (*Cordia myxa* L. Fruit (Assyrian Plum) and *Phoenix dactylifera* L. Biowaste (Date Pit)) on the Physicochemical, Microstructural, Nutritional, and Sensorial Properties of Gluten-Free Biscuits

**DOI:** 10.3390/foods11152346

**Published:** 2022-08-05

**Authors:** Syed Muhammad Ghufran Saeed, Syed Arsalan Ali, Khizra Faheem, Rashida Ali, Angelo Maria Giuffrè

**Affiliations:** 1Department of Food Science & Technology, University of Karachi, Karachi 75270, Pakistan; 2English Biscuits Manufacturer Private Limited, Karachi 74900, Pakistan; 3Dipartimento di Agricoltura, Università degli Studi Mediterranea di Reggio Calabria, Contrada Melissari, 89124 Reggio Calabria, Italy

**Keywords:** bakery, bioactive compounds, gluten-free biscuits, pasting properties, scanning electron microscopy, sensory analysis

## Abstract

The gluten-free products available on the markets are deficient in bioactive compounds and high in cost. The present study is designed to develop gluten-free biscuits with enhanced nutritional properties. The gluten-free biscuits are formulated with rice flour (RF) incorporated with Assyrian plum fruit flour (APF) and bio-waste date-pit flour (DPF) according to the following ratios; RF:DPF:APF (100:0:0)/T0, (90:5:5)/T1, (80:10:10)/T2, and (70:15:15)/T3. The results demonstrate that flour blends with different concentrations of APF and DPF incorporated in RF have high contents of protein, damaged starch, crude fiber, ash, phytochemicals, and antioxidants in contrast to 100% RF, which shows the lowest values for all these parameters. The pasting properties of the flour blends reveals that the values of peak, final, breakdown, and setback viscosities reduce from T1 to T3. Similarly, a differential scanning calorimeter reveals that the phase transition temperature of the flour blends decreases with the increasing amylose content. Moreover, the scanning electron microscopy of the biscuit samples shows a positive contribution of APF and DPF for the development of the desired compactness of the structure due to the leaching of amylose content from the starch. The total phenol content (TPC) and total flavonoid content (TFC) increase from 38.43 to 132.20 mg GAE/100 g DW and 18.67 to 87.27 mg CE/100 g DW, respectively. Similarly, the antioxidant activities of biscuits improved. The protein and fiber contents of the biscuits increased from 10.20 to 14.73% and 0.69 to 12.25%, respectively. The biscuits prepared from T3 resulted in a firmer texture with a reduced spread ratio. However, the formulation of T1 and T2 biscuit samples contributed to desirable physical and sensory properties. Therefore, the addition of DPF and APF to RF is a sustainable way to make gluten-free biscuits as they provide adequate amylose, damaged starch, and fiber content to overcome the essential role of gluten in the baked product with nutraceutical properties.

## 1. Introduction

In certain individuals, gluten consumption, particularly prolamin, which is a major component of gluten, can trigger an autoimmune disorder, i.e., celiac disease, mainly characterized by the disruption in the gut microbiome and causes malnourishment. The gluten protein is present in wheat, rye, and barley, which are causative agents of celiac disease [1]. In recent decades, celiac disease was considered to be an uncommon disease and was thought to only affect children and people in Europe. However, in the last decade, the prevalence of celiac disease has been found not only in Europe, but also in North Africa, the Middle East, India, and Pakistan [2]. The intake of a gluten-free diet is the only effective way to treat celiac disease. Generally, gluten-free products are high in cost compared to gluten-containing food items. Furthermore, despite the growing awareness and demand for healthy food products, the nutritional status of gluten-free products remains questionable due to the poor nutritional profile of the ingredients utilized for producing them [3]. Moreover, gluten-free baked foods often have a higher glycemic index [3]. Therefore, these food products can cause serious problems for celiac patients also suffering from common metabolic disorders, such as diabetes and obesity [3].

Biscuits are ready-to-eat, inexpensive, and convenient food items with a longer shelf life [4]. Celiac disease patients generally prefer crackers and biscuits as one of the major sources of carbohydrates in their daily diet compared to healthy individuals [5]. Hence, biscuits could be considered as an ideal food product to be improved by the utilization of innovative natural ingredients, which result in the development of healthier and more nutritious products for individuals following a gluten-free diet. Numerous studies have been conducted to enhance the nutritional properties of gluten-free flour by the incorporation of legumes flour, dietary fibers, fruits, and vegetable flours for the preparation of gluten-free baked products [6]. Furthermore, Giuberti et al. [7] in their research added moringa leaf powder in the preparation of gluten-free biscuits to enhance the nutritional and physicochemical properties, while rice and bean flours were utilized by Wesley et al. [8] for the production of gluten-free biscuits as a good source of fiber, minerals, and protein. In another study, rice flour with potato starch and locust bean gum as gluten-free ingredients were used in baked products [9]. Moreover, other gluten-free ingredients, including maize [10], oat [11], sweet potato [12], water chestnut [13], peanut-pearl millet composite [14], and even infant biscuit formulas containing rice and chickpea flours [15], were studied by researchers to learn more about the desirability of food selection among the various classes of consumers [15]. In the present study, rice flour is used as the major ingredient for the preparation of gluten-free biscuits for a desirable palatable feel [16], whereas the studies mentioned above utilized pseudocereal and legume flours in the manufacturing of gluten-free biscuits, which can only partially fulfill the desire of satiety.

About one-half of the world’s population consume rice as a staple food item [17]. Rice is mainly cultivated in Southeast Asia, Bangladesh, Japan, China, and India [17]. *Oryza sativa* L. is the common species of rice that grows in Asia. High-energy rice cereal contains about 80% carbohydrates, is low in fat, and contains amino acids, mainly aspartic and glutamic acid. White rice is generally liked by consumers because of its appearance, taste, shelf life, palatability, ease of cooking, and hypoallergenic properties [17]. However, the deficiency of bioactive compounds and fiber content may be overcome by the supplementation of rice flour with the flours of legumes, fruits, and vegetables to make it functionally and nutritionally more desirable [1,4,8]. *Cordia myxa L* (Assyrian plum) fruit has gained popularity due to its medicinal properties. Furthermore, its extract has demonstrated antioxidant, anti-cancer, anti-ulcer, and hepatoprotective properties in animal models [18]. On the other hand, the pits of *Phoenix dactylifera* (date) are a rich source of dietary fibers, and polyphenols and have a significant impact on human health with anti-microbial, anti-mutagenic, anti-diabetic, and anti-carcinogenic properties [19]. Date pits represent 15% of the weight of date fruit and are considered as valuable bio-waste [20]. Moreover, the Assyrian plum and date pit represent a good source of minerals, vitamins, dietary fibers, essential amino acids, and proteins, while being low in fat and sugar contents [18,19]. Therefore, the utilization of rice flour in biscuits as a low-cost gluten-free flour incorporated with Assyrian plum fruit and date pit flours could be a promising natural blend of flours with bioactive compounds. 

The nutritional and physicochemical roles of Assyrian plum fruit and date pit flours in the formulation of gluten-free rice-based biscuits have not been explored previously. Therefore, the goal of the present study is to develop nutritionally enriched gluten-free biscuits in terms of dietary fiber, antioxidants, and bioactive compounds providing the above-mentioned properties. 

## 2. Materials and Methods

### 2.1. The Collection of Raw Material 

Sugar, eggs, salt, and butter were purchased from a local market in Karachi. Non-waxy rice flour (IRRI-6) was provided by Graib Sons Private Limited. The Sulop Chemicals provided the milk and baking powders. All the chemicals used in this study were of analytical grade and purchased from Scharlau, Dae Jung, and Merck. 

### 2.2. Preparation of the Assyrian Plum Fruit and Date Pit Flours 

The date pits were provided by Khairpur’s Date Industry, Pakistan, and the Assyrian plum fruits were freshly picked from the orchards of Karachi, Pakistan. The date pits and Assyrian plum were cleaned using distilled water. A hot-air oven (Bionics Scientific, BST/HAO-1124) was used to dry the date pits at the temperature of 50 °C for 2 h. The pretreatment was given to cleaned date pits prior to milling in order to deactivate the polyphenol oxidase (PPO), which is responsible for the degradation of bioactive compounds during processing [19,20].The date pits were heated to 88 °C for 15 s in a hot-air oven [19,20]. The date pits were ground in a laboratory cutter miller (3100 Perten Instruments). Assyrian plum fruits were manually deseeded and cut into thin slices, and finally oven dried (60 °C for 8 h) and milled in a similar manner as reported above. The obtained date pit flour (DPF) and Assyrian plum flour samples were sieved separately to acquire a uniform particle size of 60 µm.

### 2.3. Preparation of the Flour Blends

The DPF and APF were added to rice flour (RF) in the following proportions (DPF: APF: RF) 100:0:0, 100:5:5, 100:10:10, and 100:15:15. Furthermore, these treatment (T) levels were indicated as T0, T1, T2, and T3, respectively. These flour blends were characterized for proximate composition, functional, pasting, thermal, antioxidants, and bioactive compound properties.

### 2.4. Characterization of the Flour Blends

All the analyses were performed in triplicates.

#### 2.4.1. Proximate Composition of the Flour Blends

The moisture (method 44–40), amylose (method 61–03), total starch (method 76–13), damaged starch (method 76–31), protein (nitrogen-converting factor of 6.25), ash, crude fiber, and lipid content of the RF, DPF, APF, and their blends were examined according to the AACC methods [21].

#### 2.4.2. Functional Properties of the Flour Blends

The oil absorption capacity (OAC) and water absorption capacity of the DPF, APF, RF, and their blends were determined, as mentioned by Ahn et al. [22].

#### 2.4.3. Pasting Behavior of the Flour Blends

The viscosity behavior of the flour blends at different temperatures and time combinations were determined by Brabender^®^ Micro-Visco-Amylograph (OHG Duisburg, Germany), according to the AACC method 22–10 [21]. The parameters noted were gelatinization time (min), pasting temperature (°C), and viscosities (peak, final, breakdown, and setback) (BU).

#### 2.4.4. Differential Scanning Calorimetry of the Flour Blends

The thermal properties of the flour samples were determined by using a differential scanning calorimeter (DSC) (Q 2000, TA instruments, New Castle, DE, USA) [23]. Briefly, the flour sample (4 mg) and 10 μL of distilled water were mixed in an aluminum pan and kept at 30 °C for 24 h. Thermal scanning of the sealed sample was performed from 30 to 120 °C at the heating rate of 10 °C /min. Thermal transitions, i.e., gelatinization enthalpy (Δ*H*), gelatinization conclusion temperature (*Tc*), gelatinization peak temperature, and onset temperature (*To*), were determined by utilizing Universal Analysis Software (2000, TA Instruments, New Castle, DE, USA).

#### 2.4.5. Antioxidant Activity of the Flour Blends

The antioxidant activity of the flour blends was determined by using the methanolic extract of samples (60, 125, 185, and 250 mg/mL). The extract was vortexed for 2 min and sonicated for 30 min in an Ultrasonicator (SONOREX RK 31, Bandelin Electronic KG, Berlin, Germany). The content was centrifuged at 3500 rpm for 5 min and the supernatants were stored as final extracts for further analysis [24].

##### Radical Scavenging Activity by 2,2-Diphenyl-1-Picrylhydrazyl (DPPH)

The radical scavenging activity of extracts of the flour blends was determined by the method of Fan et al. [25]. The DPPH solution was prepared by solubilizing 33.90 mg of DPPH in 100 mL of methyl alcohol. The prepared methanolic extract (1 mL) of flour blends was allowed to react with the DPPH solution (1 mL) for 30 min in the dark. The absorbance of the reaction mixture was recorded at 517 nm using a spectrophotometer (Perkin Elmer, Lambda 25, and UV-Vis Spectrophotometer). The % scavenging activity is determined as follows:$$\mathrm{Scavenging}\;\mathrm{activity}\;\%=\frac{\mathrm{Absorbance}\;\mathrm{of}\;\mathrm{control}-\mathrm{Absorbance}\;\mathrm{of}\;\mathrm{sample}}{\mathrm{Absorbance}\;\mathrm{of}\;\mathrm{control}}\times 100$$

The IC_50_ values (mg/mL) of the flour blends were determined by the interpretation of linear regression analysis.

##### Ferric/Ferricyanide (Fe^3+^) Reducing Antioxidant Power (FRAP)

The FRAP of the flour blends was determined, as described by Gawlik et al. [26]. The resulting reaction mixture demonstrated the development of the Perl’s Prussian color, which was measured at an absorbance of 700 nm. An increase in the intensity of color formation resulted in the increase in absorbance values, which indicated a greater FRAP.

#### 2.4.6. Bioactive Compounds in the Flour Blends

##### Total Phenolic Content

Folin–Ciocalteu reagent was used to determine the total phenolic content (TPC) of the samples [27]. The absorbance of the resulting reaction mixture was measured against a blank at 765 nm. The results were reported as milligram gallic acid equivalent/100 g (mg GAE/100 g) of the extract on a dry weight (DW) basis using a standard calibration curve.

##### Total Flavonoid Content

The total flavonoid content (TFC) of the flour blends was measured using the experimental technique of Saeed et al. [28]. The absorbance was spectrophotometrically determined at 510 nm. The extract was measured in milligram catechin equivalent/100 g (mg CE/100 g) on a dry weight (DW) basis.

### 2.5. Gluten-Free Biscuit Preparation

Biscuit samples were prepared by using RF, DPF, and APF in various combinations according to the method of Ali et al. [29] (Table 1). Butter and icing sugar were mixed for 3 min in a dough mixer (Kenwood KVL4100W, UK), and then eggs were added to the mixer and mixed for 5 min. Furthermore, the flour blends, baking powder, and milk powder were added to the dough mixer, and the mixture was mixed along with the required level of water for 3 min. The dough was wrapped in plastic wrap and sheeted to a uniform thickness of 0.9 cm. The biscuits were cut into a circular shape by using a biscuits cutter with a fixed diameter of 3.9 cm, followed by baking in a preheated oven (Anex, AG-3079, China) at 160 °C for 30 min, followed by cooling at an ambient temperature (25 °C). Furthermore, the biscuits were kept in air-tight containers until further use. About 27 biscuits were baked per batch, and each batch was prepared in triplicates.

### 2.6. Micromorphology of the Biscuits

The images of samples were taken at 500 X by Analytical Scanning Electron Microscope, JSM-6380 A, JEOL, Dearborn, MI, USA, available at the Centralized Science Laboratory in the University of Karachi.

### 2.7. Evaluation of the Quality Attributes of the Gluten-Free Biscuits

The following analyses were performed in triplicates.

#### 2.7.1. Antioxidant Activities and Bioactive Compounds

The same methods as those described above were used to evaluate the DPPH, FRAP, TPC, and TFC in order to evaluate the antioxidant capacity and bioactive components of biscuit samples [25,26,27,28]. Additionally, biscuit extracts (60, 125, 185, and 250 mg/mL) were prepared in the same manner as the flour blend extracts described above [24].

#### 2.7.2. Dimensional Analysis

A Vernier caliper was used for measuring the diameter and thickness of the biscuits. The spread ratio of the biscuits was calculated by dividing the values of the diameter/thickness [27].

#### 2.7.3. Textural Analysis

A texture analyzer (UTM, Zwick/Roell, Germany) was used to determine the breaking force (Hardness) of the biscuits by operating the three-point bending test rig technique (trigger force: 50 N, distance: 10 mm, pre-test speed: 1.0 mm/s, test speed: 5.0 mm/s, post-test speed: 10.0 mm/s, and load cell: 5 kg) [30].

#### 2.7.4. Color Analysis

The color values *L*, a*,* and *b** were determined by using an NH300 Portable Colorimeter [27]. The colorimeter was placed on four different positions on each biscuit samples, and the average value was calculated. The *L** value indicated the colors: 100 for perfect white and 0 for black, while *a** indicated (+) redness/(−) greenness and *b** (+) yellowness/(−) blueness, which were the chromaticity values.

#### 2.7.5. Nutritional Analysis

The moisture (method 44–19), protein (6.25 nitrogen-converting factor) (method 08–01), lipid (method 30–25), crude fiber (method 32–10), and ash (method 46–10) contents of the biscuit samples were determined according to the AACC method [21]. The total carbohydrate content was estimated by the difference method, i.e., carbohydrate = 100 − (crude fiber% + ash% + fat% + protein% + moisture %). Furthermore, the calories were calculated by the Atwater general factor system: 4 kcal/g for carbohydrate and protein, and 9 kcal/g for lipid.

#### 2.7.6. Sensory Evaluation

The sensory evaluation of the biscuit samples was conducted in the sensory laboratory according to the method of Ali et al. [29]. The samples of biscuits were judged by 95 semi-trained panelists (consisting of males and females of age groups ranging from 25 to 45 years) from the Department of Food Science & Technology, University of Karachi (Pakistan). The participants were trained in the baking laboratory by using the sensory profiles methodology with the commercial biscuits and prototypes [31]. The semi-trained panelists used a 9-point hedonic scale (9—like extremely, 8—like very much, 7—like moderately, 6—like slightly, 5—neither like nor dislike, 4—dislike slightly, 3—dislike moderately, 2—dislike very much, 1—dislike extremely) for evaluating the appearance, color, taste, texture, and overall acceptability of the biscuit samples. The prepared biscuit samples were analyzed in daylight and in portable cabins within the sensory laboratory. One full biscuit of each reference was consumed, which was kept at an ambient temperature (25 °C). The evaluators were given water to drink between the samples to clean their mouths. The samples were not replicated. However, the control biscuit was presented two times in the analyses, randomly between the other samples, to test the reliability of the results. 

### 2.8. Statistical Analysis

Statistical Package for the Social Sciences (SPSS) software (Version 17.0. Inc., Chicago, IL, USA) was used to analyze the data by Analysis of Variance (ANOVA). Duncan’s multiple range tests were used to determine any significant differences between the treatments at *p* ≤ 0.05. A correlation test was performed to identify the nature of correlations between antioxidant activities (DPPH and FRAP) and bioactive compounds (TPC and TFC) by using OriginPro 2022 software (Version 9.90, Northampton, MA, USA). To display the results, correlation graphs (Correlograms) were generated. Principle Component Analysis (PCA) was used to correlate the experimental values of proximate composition, functional, pasting, thermal and antioxidants properties, and bioactive compounds of the flour blends. Similarly, PCA was also applied to the data of the biscuit samples (antioxidants, bioactive compounds, color values, dimensional, textural, nutritional, and sensory properties) by XLSTAT (Addinsoft, 2020; Version 2019.2.1, Paris, France).

## 3. Results and Discussion

### 3.1. Physicochemical Properties

The proximate compositions of RF, DPF, APF, and their blends, presented in Table 2, show that the level of DPF and APF incorporation enhances the moisture and ash contents of the flour blends. The increased moisture content depicts the presence of hydrophilic components, such as fiber, starch, and protein in the DPF and APF, which facilitates increased water absorption. These observations were confirmed further by the findings of Hussain et al. [10], which reported that the high concentration of amylose content in *Cordia myxa* promotes increased water absorption. One of the main advantages of the addition of DPF in the flour blends was the high content of proteins and fibers [32], which are well-known sources used to enhance the nutritional profile of food products. The highest increase in protein content was observed in T3, which was followed by T2 and T1. In addition, the fiber content was also higher in all the flour blends, except T0. Furthermore, fiber is one of the key components of DPF (48.32 %) and APF (16.56 %) (Table 2). Despite the health benefits associated with fiber consumption, which include metabolic parameters, microbiota composition, and metabolite synthesis, fiber consumption remains low in Western countries [33]. As a result, the enhancement of food products with fiber content is a hot topic in the food industry, and it creates an equal opportunity for food reformulation [34]. A similar observation of an increase in protein and fiber contents was reported by Saeed et al. [32] when different ratios of DPF were incorporated in wheat flour. Lipid content is another essential factor in the proximate analysis. In this sense, T1, T2, and T3 showed an increase in lipid content due to the higher concentration of lipids in DPF and APF compared to RF. Date pits exhibit excellent lipid profiles due to their polyunsaturated fatty acid content, such as α-linolenic acid, which is considered as a biological precursor of eicosapentaenoic and docosahexaenoic acids [35]. The grinding or milling of cereal grains damages the starch content. The concentration of damaged starch depends on the processing factors of milling and the mechanical forces, i.e., dry milling produces more damaged starch than wet and semi-wet milling [36]. Moreover, wet milling consumes a lot of energy and water, which not only wastes water but also pollutes the environment. Furthermore, the soaking of grains for an extended period of time is difficult without causing the growth of bacteria, which has a direct influence on the safety and quality of food [37]. In comparison, dry milling is the process of producing flour in a dry environment by using various grinding equipment. On the other hand, the technology of dry milling consumes less energy without the wastage of water, and retains the nutrients of milled products, which are usually decreased or lost during the pre-treatments of wet milling [38]. In this study, dry milling was performed due to the above-mentioned reasons. The lowest damaged starch content was observed in T0 (7.92%) and the highest in T3 (8.76%).

### 3.2. Functional Properties of the Flour Blends

The functional properties of flour are strongly affected by the composition of the flour and may have a direct impact on the final product’s characteristics. The functional properties of rice flour blends utilized for biscuit manufacturing are presented in Table 2. The water absorption capacity of flour blends increased with the addition of DPF and APF, T0 showed the lowest value (138.32 %), whereas the highest value (226.71%) was observed for T3. The increased WAC depicted the increased quantity of fibers, proteins, and other hydrophilic components in the flour blends [28]. The increased WAC of the flour blends was correlated with the results of the increased moisture content, as reported in Table 2, due to similar reasons. Furthermore, the variations in the WAC may also depend on the nature of amino acids (hydrophilic or hydrophobic) and their surface distribution in the food system [39]. Similarly, the presence of non-polar side chains of amino acids in the flour blends affects the oil absorption capacity (OAC). The data (Table 2) show that higher concentrations of DPF and APF in rice flour result in a higher OAC. A similar observation of increases in the OAC was observed when various levels of DPF were incorporated in wheat flour [32]. Food ingredients with a high OAC can assist in maintaining a healthy blood lipid profile and body weight by reducing the lipid absorption in blood from the ingested meals in the gastrointestinal system [40]. Moreover, the OAC is an important property of flour used in baked foods, which has a positive impact on flavor retention, mouthfeel, shelf life, textural profile, and palatability [39]. The literature revealed that greater sensory scores of buckwheat gluten-free biscuits were observed due to the increased OAC of buckwheat flour [39]. Furthermore, *Cordia myxa* contains stable polysaccharides, which play an important role in enhancing these functional properties through their β glyosidic linkage bonds [41]. Therefore, these flour blends with a high WAC and OAC can be used as functional ingredients for the production of healthier food products.

### 3.3. Pasting Properties of the Flour Blends

The viscometric properties of the flour blends have a direct impact on their textural and sensory properties, i.e., the mouthfeel of food products. Samples T1 and T2 showed higher values of peak and final viscosities, while T3 showed lower values (Table 3). Moreover, T3 had a lower amount of total starch content than then other flour blends (Table 2), thereby contributing to a decrease in peak and maximum viscosities during heating [42]. However, the peak viscosity decreased with the incorporation levels of DPF and APF. The decrease in viscosity was due to the increased concentration of dietary fiber and protein content in the flour blends that bound with the water and restricted the granular swelling of starch components [32]. Furthermore, setback viscosity was highly dependent on the amylose content of the starch. The amylose content of T0 was lower than T1, T2, and T3 (Table 2), which could explain the lower setback viscosity of the flour blends incorporated with DPF and APF. The data apparently reveal that a higher amylose content contributes to a decrease in viscosity and increase in the gelatinization time as amylose contributes to the availability of more amorphous regions. In addition, the lower setback viscosities of the flour blends suggest a lower degree of starch retrogradation, which contributes to the longer shelf life of baked products made from these flour blends. Similarly, Mahmood et al. [43] observed a decline in setback viscosity when *Cordia myxa* fruit gum was used in a food system. The breakdown viscosity has a direct relation with the stability of starch molecules under high-heat and shear forces [44]. Moreover, the flour blends (T1, T2, and T3) showed a decreasing trend in breakdown viscosity, which was correlated with good paste stability and strong shearing resistance [29].

### 3.4. Thermal Properties of the Flour Blends

It was observed that the conclusion temperature (*Tc*), peak temperature, and onset temperature (*To*) of gelatinization decreased with the increase in the concentration of APF and DPF in the RF (Table 3). Furthermore, the decreasing trend of temperatures was correlated with the increased level of amylose content in the flour blends (Table 2). A similar observation of a reduction in the values of *To*, *Tp*, and *Tc* was reported by Ye et al. [23] for rice flour with an increased concentration of amylose content. The results suggest that the fortification of RF by APF and DPF in biscuit samples delays the gelatinization of starch granules, as higher values of *To*, *Tp*, and *Tc* can be observed in relation to samples without fortification, i.e., plain RF [45]. The higher temperature detected for composite flour blends than RF was due to the fact that APF and DPF have different starch origins, which might have higher gelatinization temperatures than rice starch [46]. The literature reveals that a positive correlation exists between *Tp*, *Tc*, and crystallinity in rice flour [47]. Moreover, a greater degree of crystallinity of flour caused an increment in the transition temperature due to the greater stability and resistance of starch granules for the rate of gelatinization [47]. Therefore, the decreasing values of the transition temperature of the flour blends might be associated with the presence of a lower degree of crystallinity with the increase in the incorporation levels of APF and DPF in RF. Thus, APF and DPF are known to have amorphous characteristics [48]. Furthermore, the enthalpy of gelatinization (Δ*H*) gradually increased with the level of fortification from 6.11 to 6.82 J/g. The increase in Δ*H* was attributed to the presence of hydrophilic non-starch components in APF and DPF, such as soluble fibers and proteins, which competed for the available molecules of water and reduced the hydration and mobility of starch polymers, and ultimately resulted in a higher energy requirement for melting [49]. Furthermore, the findings or the thermal properties of the flour blends correlated with the observation of pasting characteristics, i.e., the decline in breakdown viscosity with respect to the addition of APF, and DPF in RF also predicted the thermal stability of these flour blends due to increased shearing resistance (Table 3).

### 3.5. Scanning Electron Micrograph of the Biscuits

The scanning electron micrographs of biscuit samples are presented in Figure 1. The overlapping of starch granules in the biscuit samples is visualized in Figure 1c, which was formulated from 100% RF (T0). The incorporation of DPF and APF in biscuit samples resulted in the development of starch and protein interactions. Protein bodies of DPF and APF can be observed, along with the starch granules of RF, which seem to be embedded in the thin matrix of protein fibrils (Figure 1d,g). Moreover, smooth surfaces are present in Figure 1d,g, which were formed due to the formation of the starch–lipid complex (Figure 1d,g). The starch and protein interactions at higher concentrations of DPF and APF in the biscuit samples (T2 and T3) are shown in Figure 1g,h, which caused the granules to hold steam generated during baking. The absence of a gluten network is compensated by the interactions of starch to starch, protein to starch, and protein to protein. However, a higher concentration of DPF and APF in the biscuit samples resulted in the development of irregular patterns (Figure 1h). The compactness and irregularities in the structure of T3 (Figure 1h) was responsible for the harder texture of the biscuit samples (Table 4).The increased concentration of fiber restricted the swelling of starch granules, which may be due to the protecting effect of the components of fiber that decreases the rate of heat transfer during baking. Another reason could be the competition for the available water among the starch and fiber components. Similar observations were reported by Tavares et al. [50], where okara, rice flour, broken rice flour, and rice bran were used for the manufacturing of gluten-free biscuits.

### 3.6. Antioxidant Activity

The antioxidant activities of the flour blends and their biscuit samples are presented in Table 4. The DPPH radical scavenging activity of the flour blends increased with an increase in the concentration of DPF and APF incorporated in RF. The highest radical scavenges activity was observed for T3 flour blends and their biscuit samples and depicted the lowest IC_50_ values of 96.04 mg/mL and 80.04 mg/mL, respectively, while the highest IC_50_ value was witnessed by 100% RF (378.21 mg/mL) and their biscuits (360.10 mg/mL).

A similar trend of antioxidant activity was observed for the ferric reducing antioxidant power (FRAP) as mentioned for DPPH inhibition activity (Table 4). The FRAP increased significantly (*p* ≤ 0.05) with an increase in the levels of DPF and APF in the flour and biscuit samples. The biscuits produced from T3 flour blends showed the highest FRAP (i.e., lower IC_50_ values) compared to T0. The antioxidant potential of flour blends and biscuit samples was due to the presence of antioxidants and bioactive compounds in DPF and APF. Furthermore, Habib et al. [51] and El-Massry et al. [52] suggested that epicatechin, catechin, gentisic acid, and chrysin were the phytochemicals present in DPF and APF with a potent antioxidant potential. The mechanism responsible for the antioxidant activity of the samples was linked with the availability of antioxidants present in DPF and APF, which function as reducing agents and neutralize the formation of the free-radical chain reaction by the donation of electrons, which ultimately restricting the synthesis of peroxides [28]. Furthermore, due to the formation of melanoidins (antioxidants) during the baking process, the antioxidant activity of the biscuit samples was increased [53]. A similar observation of the increase in DPPH inhibition and FRAP for flour blends and biscuit samples was depicted by Saeed et al. [32] when various levels of DPF were incorporated in wheat flour to improve the antioxidant potential.

### 3.7. Total Phenolic Content (TPC)

The nutraceutical value of any substance can be correlated with the presence of polyphenolic compounds, which are bioactive compounds [28] (Saeed et al., 2020). The TPC increased with the incorporation level of DPF and APF in the flour blends. However, a considerable decrease in the TPC of the biscuit samples was observed, which was not compensated by the formation of melanoidins during baking. Additionally, the thermal degradation of polyphenols and the decarboxylation of phenolic acids that occurred during the heat treatment was the cause of the reduction in the TPC of the biscuit samples [54]. A greater reduction in the TPC was in the biscuit samples formulated from T0, which showed a 9.637% loss of TPC after baking, and the lowest was observed for T3 biscuits, which showed a 4.32% loss of TPC after baking. Despite a greater decrease in TPC after baking, T1, T2, and T3 showed the highest values of TPC compared to T0 (control biscuits). Saeed et al. [32] depicted a similar decrease in the TPC of biscuits made from different concentrations of DPF.

### 3.8. Total Flavonoid Content (TFC)

Flavonoids are one of the broad groups of phytochemicals with antioxidant and radical scavenging properties [55]. From the data reported in Table 4, it can be observed that the higher concentration of DPF and APF increases the TFC in the flour blends and their gluten-free biscuits. Similarly, gluten-free biscuits prepared from T3 showed the highest value of TFC, while the lowest value was observed for T0. Moreover, the literature reveals that date pits exhibit higher concentrations of flavonoid-containing compounds, such as rutin, catechin, and epicatechin, than the whole date fruit [56]. The presence of these flavonoid compounds in flour blends and biscuit samples was responsible for the increase in the TFC. Furthermore, during the baking process, the TFC increased significantly (*p* ≤ 0.05) compared to flour blends. The maximum increase in TFC was identified in biscuit samples prepared from T3, which showed a 40.41% increase in TFC during baking, and a minimum increase was observed for T1 biscuits, which showed a 36.43% increase. The observed variations in the values of TFC may be due to the formation of brown-colored pigments, i.e., melanoidins, which are the byproducts of the Maillard reaction generally developed during the baking of biscuits [57] (Cervini et al., 2021). A similar increase in TFC was observed by Pasqualone et al. [58] for biscuit samples prepared from purple wheat, and suggested that the increase was due to the formation of volatile complexes generated by the plant during heat treatment, i.e., baking. Platat et al. [55] also depicted the increase in the TFC of pita bread incorporated with DPF as a replacement of wheat flour.

### 3.9. Correlation Studies of Antioxidants with Bioactive Compounds

The coefficient values of the Pearson correlation and their levels of significance analyzed between the TPC, TFC, DPPH-IC_50_, and FRAP-IC_50_ are illustrated in a correlation plot matrix (Figure 2a,b), respectively. The positive correlations are presented in red, while negative correlations are presented in blue. The correlation coefficients were inversely proportional to the size of the ellipse, i.e., the larger the size, the more negative the correlation. Correlations with non-significant differences (*p* > 0.05) were not included in the correlogram (Figure 2b). The incorporation of DPF and APF in the flour blends and their resultant biscuit samples depicted the positive correlation between them, with respect to the values of DPPH-IC_50_ and FRAP-IC_50_. Moreover, strong and positive correlations were expected due to the fact that these antioxidant assays work via the same mechanism, i.e., the donation of a single electron [32]. The lower IC_50_ values of DPPH and FRAP were associated with increased antioxidant activity. The TPC exhibited negative linear correlations with IC_50_ values of DPPH and FRAP for flour blends and biscuits. Similarly, TFC was directly related to the antioxidant activity of flour blends and biscuit samples, as shown by the negative linear correlation between TFC and IC_50_ values of DPPH and FRAP. These observations suggested that the reduction in IC_50_ of DPPH and the IC_50_ of FRAP potentially led to an increment in the values of TPC and TFC, implying the fact that the DPF and APF have a strong antioxidant potential. Moreover, TFC and TPC established a positive correlation between them, with flavonoids belonging to phenolic compounds relying on the antioxidant mechanisms of DPF and APF.

### 3.10. Physical Properties of the Biscuits

The physical properties of the biscuit samples are presented in Table 4. The diameter of the biscuit samples prepared from the flour blends of T1 and T2 was higher than those for T0 and T3. Similarly, the spread ratio of biscuit samples increased with the level of incorporation of DPF and APF. However, the T3 biscuit sample showed a decline in the spread ratio, which may have been due to the presence of a higher concentration of fibers as compared to other flour blends [28]. Generally, an increase in the spread ratio is desirable in biscuit manufacturing [32]. The textural values of biscuit samples, which is considered as a force required to break the biscuit, increased significantly (*p* ≤ 0.05) with the level of DPF and APF. From Table 4, it can be observed that T1 and T2 have a lower hardness value than T3. The T3 biscuit samples showed the maximum value for the breaking force, which was due to the increased quantity of starch and dietary fiber contents. Moreover, a higher amount of dietary fiber and polysaccharides, i.e., starch, resulted in a significant increase in the compactness of structure, which increased the hardness value of the gluten-free biscuits [57]. Moreover, Dogan and Meral [9] reported that increased levels of starch content caused the firmer texture of the baked products. A similar observation was reported by Giuberti et al. [7] when gluten-free biscuits were formulated by the utilization of moringa leaf powder. Biscuit color was also affected by the rate of DPF and APF additions. The *L*, a**, and *b** values progressively decreased, increased, and increased, respectively, following the increase in DPF and APF in the biscuit samples. The lower *L** and higher a* values indicated that the T1, T2, and T3 samples were darker than the T0 samples. Furthermore, the DPF and APF were darker in color (brownish) than rice flour due to the fact that they are rich in polyphenolics and mineral content, which affected their color. Moreover, biscuit samples (T1, T2, and T3) with higher protein contents promoted non-enzymatic browning via Maillard reactions [32]. Furthermore, the browning effect of DPF and APF has been considered as a positive factor because one of the negative attributes of many gluten-free bakery products is their lighter color compared to conventional baked foods. Similar observations of color parameters were reported by Cannas et al. [1] when rice flour was substituted with quinoa flour for the development of gluten-free biscuits.

### 3.11. Nutritional Analysis of the Biscuits

The chemical composition of rice-based gluten-free biscuits with different levels of DPF and APF is presented in Table 5. The gluten-free biscuits (T1, T2, and T3) showed higher contents of protein, ash, and fiber, and a lower fat content than T0. The increase in protein, ash, and dietary fiber contents in the biscuits was due to the fact that DPF and APF contained higher amounts of protein, ash, and fiber than the rice flour (Table 2). Saeed et al. [32] also reported that DPF-containing biscuits had higher concentrations of protein, ash, and fiber than the control biscuits. In another study, Cannas et al. [1] observed that gluten-free rice-based biscuits enriched with quinoa flour exhibited a lower level of fat content due to the reduction in fat extractability by the formation of amylose–lipid complexes during baking. This might be possible in the present study since biscuits containing DPF and APF with high amylose contents (Table 2) might have developed higher levels of amylose–lipid complexes during the baking of the biscuits, which could have reduced the extraction rate of lipids. Moreover, the formation of the amylose–lipid complex may play an important role in increasing the shelf life of food products due to increased oxidative stability and the reduced retrogradation of starch components [1]. In addition, these amylose–lipid complexes are of great nutritional interest because they may have physiological effects similar to dietary fiber and resistant starch. The regular consumption of foods containing amylose–lipid complexes has been shown to reduce blood glucose levels in humans and the proliferation of colon cancer in rats [58]. From the results reported in Table 4, it can be observed that T1, T2, and T3 gluten-free biscuit samples contain high levels of polyphenols. However, In vitro carbohydrate digestibility was not considered in the present study. It may be suggested that interactions of polyphenols–carbohydrates can slow down the digestion of carbohydrates through the inhibition of digestive enzymes or the modulation of glucose uptake [59]. The moisture content of biscuit samples was not significantly (*p* ˃ 0.05) different from one another. Moreover, the literature reveals that gluten-free products are generally deficient in protein and dietary fiber contents and higher in carbohydrate and fat contents [1]. However, the present study suggested that the biscuits prepared with the addition of DPF and APF had higher nutritional qualities with higher protein, ash, and fiber contents, and were lower in kcal.

### 3.12. Sensory Analysis

The effect of the incorporation of DPF and APF on the sensorial properties of gluten-free biscuits was evaluated (Table 5) and the images of biscuit samples are illustrated in Figure 1. The sensory scores for appearance, color, taste, texture, and overall quality of gluten-free biscuits increased correspondingly to the increase in the amount of DPF and APF in the formulations. However, the sensory scores declined at higher concentrations of DPF and APF in biscuit samples (T3). The observations depicted the fact that the darker color, bitter taste, and firmer texture of biscuits (T3) were not accepted by the panelists. Furthermore, the development of bitter compounds may be correlated to the presence of high concentrations of polyphenolic compounds (Table 4) [60]. The firmness of the biscuits’ texture was due to the formation of clusters between amylopectin, DPF, and APF [43]. Moreover, the incorporation of ingredients that contained high levels of starch and fiber contents resulted in the firm texture of the biscuits due to their strong affinity for water molecules and the association of starch components with the non-starch substances, i.e., proteins and lipids, which were responsible for the development of the hard and compact structure [57]. Similarly, Kaur et al. [39] studied the sensorial characteristics of bread with the value-added attribute of *Cordia myxa*, and concluded that bread made with higher percentages of *Cordia myxia* produced a firmer texture and darker color. Another study reported the increased firmness and reduced overall acceptability of gluten-free biscuits when the concentration of resistant starch increased [57].

### 3.13. Principal Component Analysis

The differences and similarities between the samples after the incorporation of different concentrations of DPF and APF in rice flour and its biscuit samples were evaluated by PCA (Figure 3). The pasting characteristics, bioactive compounds, functional properties, and physicochemical properties of flour blends were monitored (Figure 3a,b). PC1 and PC2 cumulatively explained 99.17% of the total variation among the 22 attributes of 4 samples (Figure 3). PC1 and PC2 explained 77.48% and 21.69% variations in the data, respectively. The positive correlations between T1, T2, and T3 were found as T1 and T2 appeared closer in the bi-plot. Moreover, T3 appeared closer to T2. However, T0 and T3 appeared opposite to each other, which showed a negative correlation. The bi-plot of PC1 vs. PC2 showed that the maximum attributes were associated with PC1 (Figure 3b). A positive relationship was observed between the physicochemical attributes (protein%, fat%, crude fiber%, ash%, amylose%, damaged starch%, and total starch and moisture %), antioxidant properties (TFC and TPA), functional properties (WAC% and OAC %), pasting properties (pasting time, peak viscosity, and gelatinization temperature), and thermal properties (*To*, *Tp*, *Tc,* and Δ*H*) in PC1. Furthermore, PC2 positively correlated to setback viscosity and final viscosity. However, the breakdown viscosity negatively contributed to the variation among the flour blend samples.

Using PCA, the average values of the physical and nutritional profile of gluten-free biscuits were evaluated, and the results are shown in Figure 4a,b. Together, PC1 and PC2 accounted for 96.54% of the variance in the 16 characteristics of the 4 samples of gluten-free biscuits. It was determined that PC1’s eigenvalue was 12.45 and explained 77.87% of the variability. T0, T1, and T2 were found close to each other in the bi-plot (Figure 4b), which showed similar physical (thickness, diameter, spread ratio, and *L**) and nutritional (energy, moisture, ash, protein, carbohydrate, and fat) characteristics for these gluten-free biscuit samples. The average values of the sensory characteristics of the gluten-free biscuit samples were also analyzed by performing PCA (Figure 5a,b). The eigenvalue of PC1 was found to be >1 and accounted for 80.76% of the variation. The bi-plot (Figure 5b) clearly shows that T0, T1, and T2 are positively correlated to each other in terms of color, texture, taste, appearance, and overall acceptability. Furthermore, from Figure 5a, it can be observed that T3 significantly shows different traits compared to other biscuits samples, which clearly explains the negative correlation among them. PCA and ANOVA for sensorial parameters depicted similar results (Table 5).

## 4. Conclusions

A gluten-free diet is the only solution for individuals diagnosed with celiac disease. Generally, gluten-free products available on the market are lacking in phytochemicals and macronutrients, which can cause malnutrition. The present study was designed to develop gluten-free biscuits with enhanced nutritional profiles by incorporating DPF and APF in RF without having adverse effects on the tec + h + nological and sensorial properties. The pasting properties of flour blends depicted that the setback viscosity decreased with the increased level of incorporation, which was correlated with the reduced staling rate of the biscuits. The thermal properties revealed the higher pasting and conclusion temperature of flour blends than T0, which showed the thermal stability of flour blends. It was evident from the results that the addition of DPF and APF in the biscuits’ preparation resulted in increased DPPH radical scavenging activity, FRAP, TPC, and TFC compared to the control (T0). Furthermore, the protein and fiber contents of the biscuit samples increased. The T1 and T2 biscuit samples showed promising results with the most acceptable physical, textural, and sensorial characteristics. The study also highlights the possibility for the utilization of date seed bio-waste in gluten-free food-processing industries for cost-effectiveness, the enhancement of nutritional profile, and waste management by providing value to date-processing industries for commercialization. Therefore, in the future, the availability of gluten-free biscuits fortified with DPF and APF will expand the market/business of nutraceutical gluten-free products.

## Figures and Tables

**Figure 1 foods-11-02346-f001:**
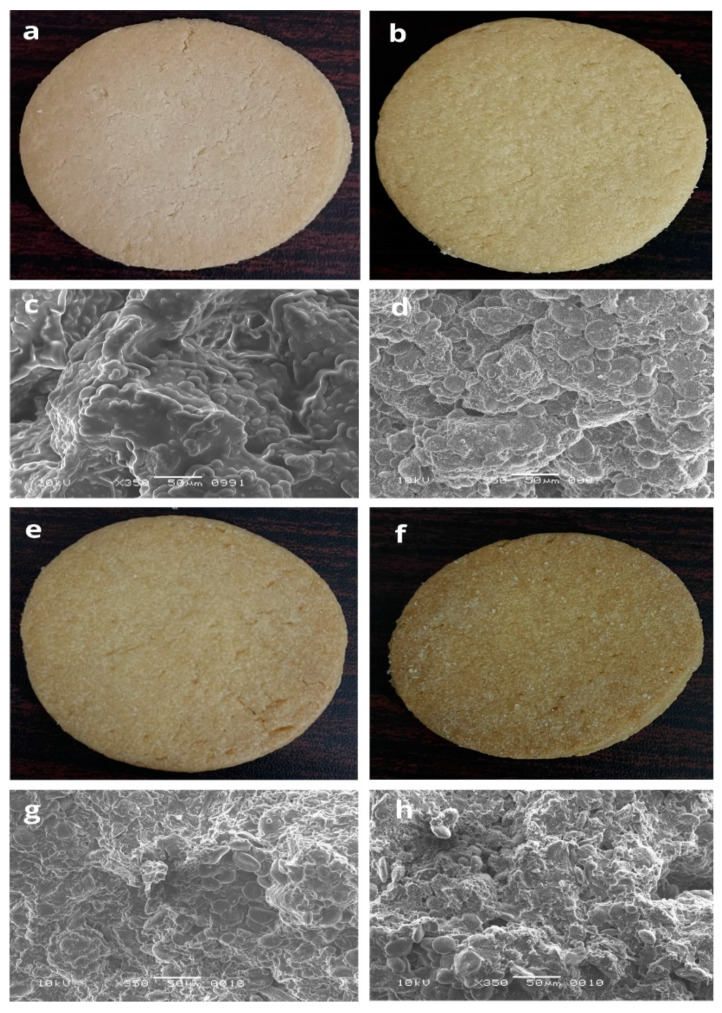
Appearance and scanning electron microscopic images (at 350× magnification) of gluten-free biscuit samples: (**a**,**c**) T0; RF:DPF:APF (100:0:0); (**b**,**d**) T1; RF:DPF:APF (90:5:5), (**e**,**g**) T2; RF:DPF:APF (80:10:10), and (**f**,**h**) T3; RF:DPF:APF (70:15:15), respectively, where T0, T1, T2, and T3 are the treatment (T) levels. Rice flour: RF, date pit flour: DPF, Assyrian plum flour: APF.

**Figure 2 foods-11-02346-f002:**
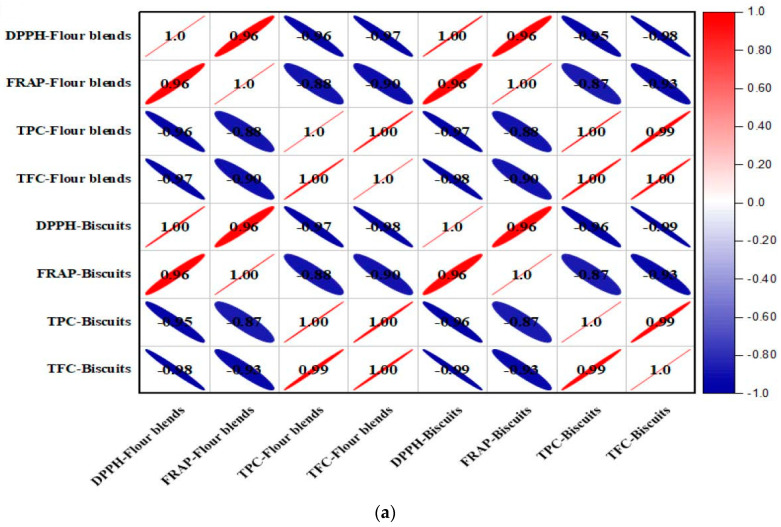
(**a**) Correlogram analysis with the obtained values of the Pearson coefficient of correlation between the bioactive compounds (total phenol content (TPC) and total flavonoid content (TFC)), and antioxidant activities (DPPH-IC_50_ and FRAP-IC_50_) of flour blends and biscuits. (**b**) Correlogram with the defined significance levels, where DPPH: 2, 2-diphenyl-1-picrylhydrazyl, FRAP: ferric reducing antioxidant power.

**Figure 3 foods-11-02346-f003:**
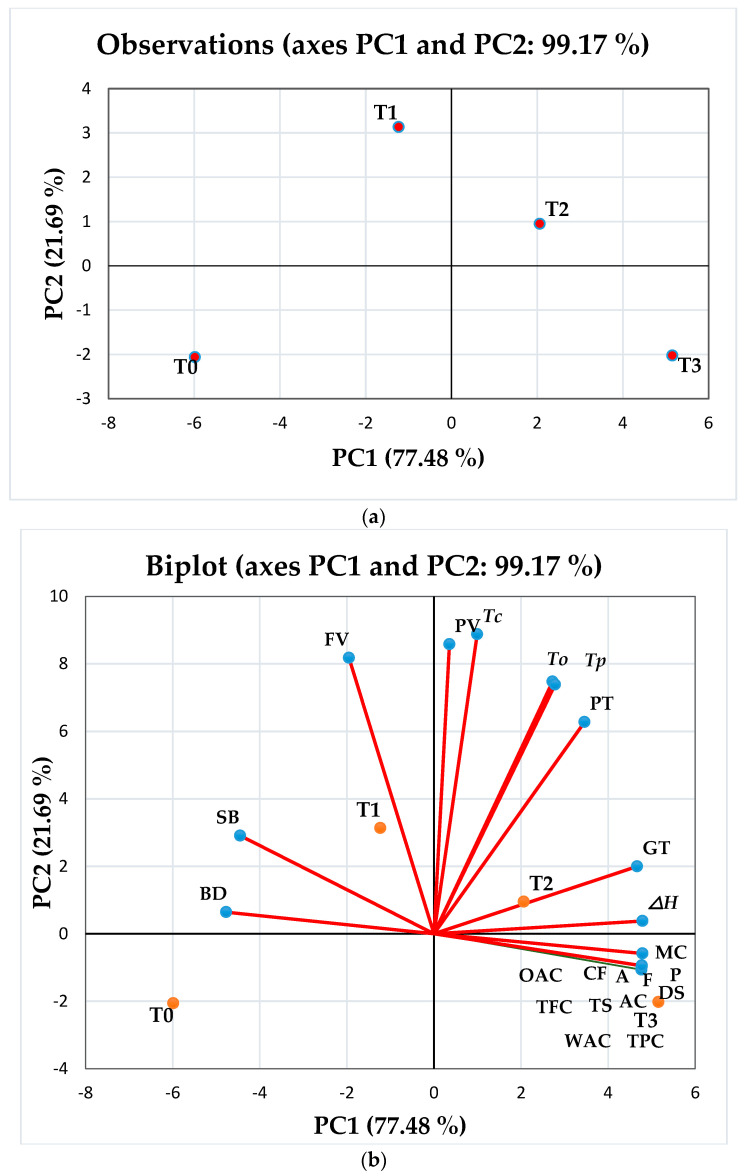
(**a**) Principal component analysis with the location of flour blend samples: T0; RF (rice flour):DPF (date pit flour):APF (Assyrian plum flour) (100:0:0), T1; RF:DPF:APF (90:5:5), T2; RF:DPF:APF (80:10:10), and T3; RF:DPF:APF (70:15:15). (**b**) Bi-plot of flour blend samples with the distribution of properties (physicochemical, functional, pasting, and bioactive compounds) in space defined by principal components 1 (PC1) and 2 (PC2), where T0, T1, T2, and T3 are the treatment (T) levels. MC; moisture content, A; ash, P; protein, CF; crude fiber, DS; damaged starch, TS; total starch, AC; amylose content, WAC; water absorption capacity, OAC; oil absorption capacity, TPC; total phenolic content, TFC; total flavonoid content, BD; breakdown viscosity, SB; setback viscosity, FV; final viscosity, PT; pasting temperature, GT; gelatinization time, *To*; onset temperature, *Tp*; peak temperature, *Tc*; conclusion temperature, Δ*H*; enthalpy of gelatinization.

**Figure 4 foods-11-02346-f004:**
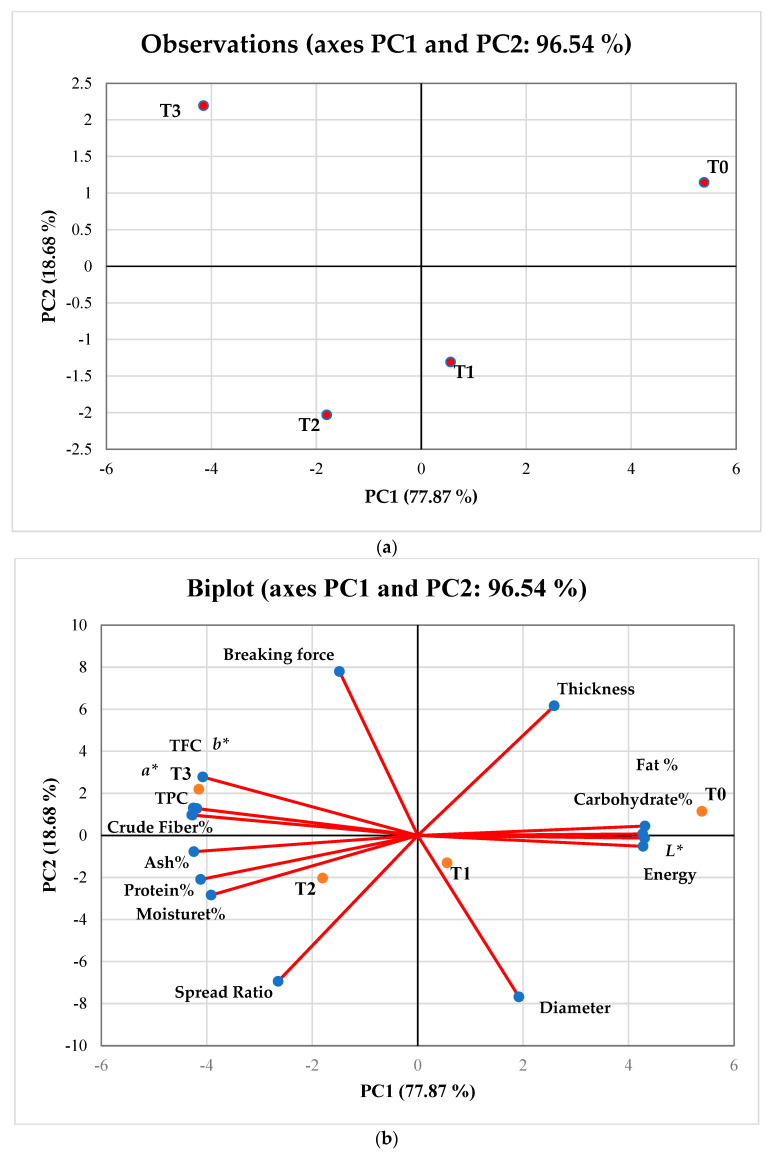
(**a**) Principal component analysis with the location of biscuit samples: T0; RF (rice flour):DPF (date pit flour):APF (Assyrian plum flour) (100:0:0), T1; RF:DPF:APF (90:5:5), T2; RF:DPF:APF (80:10:10), and T3; RF:DPF:APF (70:15:15). (**b**) Bi-plot of gluten-free biscuit samples describing the relationship between parameters (bioactive compounds, color values, nutritional, dimensional, and textural) in the space defined by principal components 1 (PC1) and 2 (PC2), where T0, T1, T2, and T3 are the treatment (T) levels. TPC; total phenolic content, TFC; total flavonoid content.

**Figure 5 foods-11-02346-f005:**
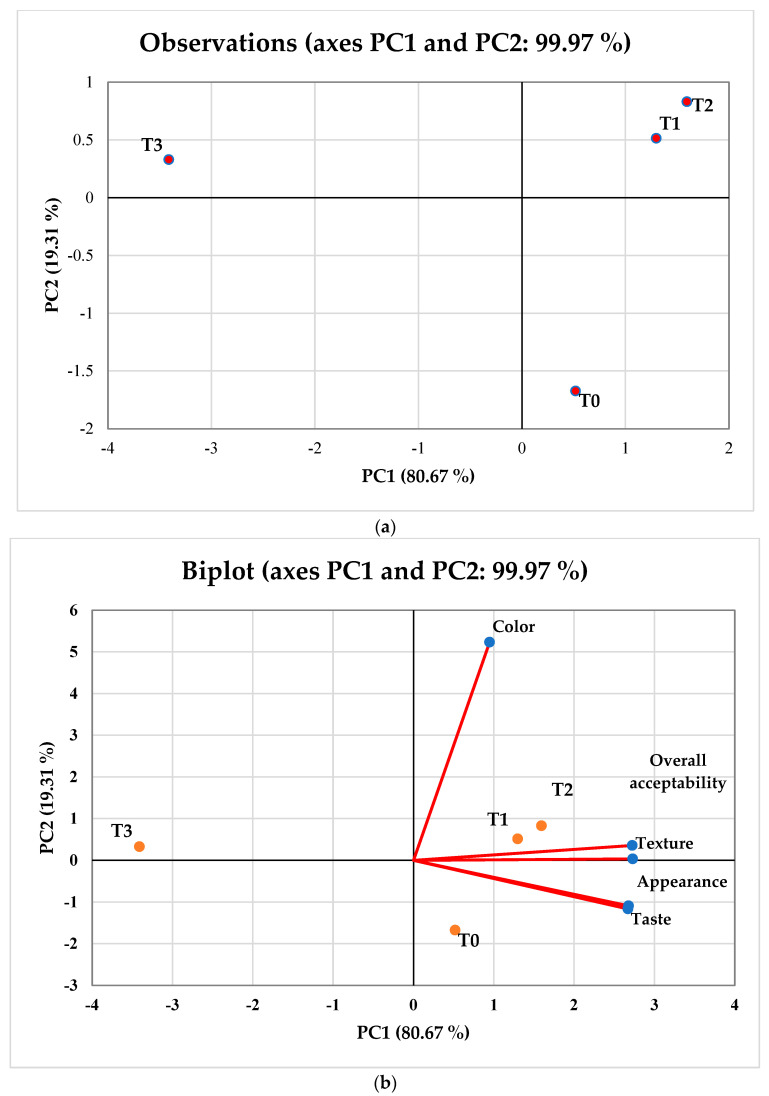
(**a**) Principal component analysis with the score plot of sensory analysis of gluten-free biscuit samples describing the variations among the samples: T0; RF (rice flour):DPF (date pit flour):APF (Assyrian plum flour) (100:0:0), T1; RF:DPF:APF (90:5:5), T2; RF:DPF:APF (80:10:10), and T3; RF:DPF:APF (70:15:15). (**b**) Bi-plot of gluten-free biscuit samples describing the relationship between parameters (bioactive compounds, color values nutritional, dimensional, and textural) in the space defined by principal component 1 (PC1) and 2 (PC2), where T0, T1, T2, and T3 are the treatment (T) levels. TPC; total phenolic content, TFC; total flavonoid content.

**Table 1 foods-11-02346-t001:** Formulations of gluten-free biscuit samples.

Ingredients	T0	T1	T2	T3
Refined rice flour (g)	100	90	80	70
Butter (g)	40	40	40	40
Sugar (g)	40	40	40	40
Egg (g)	25	25	25	25
Milk powder (g)	5	5	5	5
Baking powder (g)	1.5	1.5	1.5	1.5
Distilled water (mL)	5 ± 2	7 ± 2	9 ± 2	11 ± 2
Assyrian plum flour (g)	0	5	10	15
Date pit flour (g)	0	5	10	15

**Table 2 foods-11-02346-t002:** Proximate compositions and functional properties of rice flour (RF), date pit flour (DPF), Assyrian plum flour (APF), and their flour blends.

Proximate compositions									Functional Properties	
Sample	Moisture %	Protein %	Fat %	Ash %	Crude Fiber %	Damaged Starch %	Amylose %	Total Starch %	Water Absorption Capacity %	Oil Absorption Capacity %
T0	7.04 ± 0.11 ^a^	5.78 ± 0.12 ^a^	0.92 ± 0.01 ^a^	1.25 ± 0.03 ^a^	0.52 ± 0.02 ^a^	7.92 ± 0.14 ^f^	5.88 ± 0.12 ^a^	78.67 ± 1.10 ^f^	138.32 ± 2.01 ^a^	104.11 ± 2.02 ^b^
T1	7.16 ± 0.14 ^b^	5.94 ± 0.13 ^b^	1.20 ± 0.07 ^b^	2.14 ± 0.05 ^c^	3.71 ± 0.07 ^b^	7.41 ± 0.16 ^e^	6.05 ± 0.15 ^b^	74.50 ± 1.24 ^e^	154.01 ± 2.11 ^b^	103.19 ± 2.06 ^d^
T2	7.30 ± 0.17 ^c^	6.09 ± 0.15 ^c^	1.48 ± 0.08 ^c^	3.04 ± 0.07 ^d^	6.90 ± 0.10 ^c^	6.89 ± 0.18 ^d^	6.17 ± 0.17 ^c^	70.19 ± 1.30 ^d^	140.14 ± 2.14 ^c^	102.28 ± 2.09 ^e^
T3	7.42 ± 0.19 ^d^	6.26 ± 0.17 ^d^	1.76 ± 0.10 ^d^	3.94 ± 0.08 ^e^	10.09 ± 1.03 ^d^	6.43 ± 0.19 ^c^	6.35 ± 0.19 ^e^	66.05 ± 1.37 ^c^	185.30 ± 2.17 ^d^	101.35 ± 2.10 ^f^
DPF	7.21 ± 0.13 ^e^	6.53 ± 0.14 ^e^	5.64 ± 0.12 ^f^	2.11 ± 0.02 ^b^	48.32 ± 0.21 ^f^	2.32 ± 0.13 ^a^	6.23 ± 0.09 ^d^	32.21 ± 0.21 ^a^	268.32 ± 2.19 ^e^	117.43 ± 2.04 ^c^
APF	9.32 ± 0.10 ^f^	8.20 ± 0.11 ^f^	1.78 ± 0.04 ^e^	18.32 ± 1.03 ^f^	16.56 ± 0.17 ^e^	3.32 ± 0.11 ^b^	8.60 ± 0.10 ^f^	40.32 ± 0.36 ^b^	321.32 ± 2.21 ^f^	72.42 ± 1.01 ^a^

Each value is expressed as mean ± S.D. (*n* = 3); mean followed by different letters in the same column differs significantly (*p* ≤ 0.05). The reported results are based on a dry weight basis (db), where T0; RF:DPF:APF (100:0:0), T1; RF:DPF:APF (90:5:5), T2; RF:DPF:APF (80:10:10), and T3; RF:DPF:APF (70:15:15).

**Table 3 foods-11-02346-t003:** Pasting properties of rice flour and different ratios of date pit flour (DPF) and Assyrian plum flour (APF) incorporated in rice flour.

Pasting Properties							Thermal Properties			
Samples	Gelatinization Time (Min)	Peak Viscosity (BU)	Final Viscosity (BU)	Break Down Viscosity (BU)	Setback Viscosity (BU)	Pasting Temperature (°C)	*To* (°C)	*Tp* (°C)	*Tc* (°C)	Δ*H* (J/g)
T0	24.01 ± 0.13 ^a^	1361.02 ± 10.10 ^d^	1542.06 ± 18.11 ^d^	378.11 ± 4.13 ^d^	479.13 ± 6.10 ^d^	67.01 ± 0.23 ^a^	75.32 ± 0.16 ^d^	77.24 ± 0.24 ^d^	79.81 ± 0.33 ^d^	6.11 ± 0.01 ^a^
T1	30.11 ± 0.33 ^b^	1354.03 ± 14.11 ^c^	1528.11 ± 16.03 ^c^	367.13 ± 8.30 ^c^	472.19 ± 7.41 ^c^	71.11 ± 0.33 ^bc^	73.43 ± 0.38 ^c^	75.21 ± 0.42 ^c^	78.30 ± 0.47 ^c^	6.43 ± 0.04 ^b^
T2	33.12 ± 0.42 ^c^	1333.07 ± 15.03 ^b^	1488.10 ± 56.01 ^b^	354.09 ± 7.51 ^b^	464.21 ± 5.31 ^b^	71.12 ± 0.35 ^bc^	72.31 ± 0.35 ^b^	74.57 ± 0.37 ^b^	76.41 ± 0.40 ^b^	6.67 ± 0.06 ^c^
T3	34.01 ± 0.61 ^d^	1310.04 ± 18.12 ^a^	1442.09 ± 13.01 ^a^	346.08 ± 7.20 ^a^	442.12 ± 5.10 ^a^	70.03 ± 0.32 ^b^	70.11 ± 0.31 ^a^	72.43 ± 0.34 ^a^	73.56 ± 0.36 ^a^	6.82 ± 0.09 ^d^

Each value is expressed as mean ± S.D. (*n* = 3); mean followed by different letters in the same column differs significantly (*p* ≤ 0.05), where T0; RF:DPF:APF (100:0:0), T1; RF:DPF:APF (100:5:5), T2; RF:-DPF:APF (80:10:10), and T3; RF:DPF:APF (70:15:15). *To*: onset temperature, *Tp*: peak temperature, *Tc*: conclusion temperature, Δ*H*: enthalpy of gelatinization.

**Table 4 foods-11-02346-t004:** DPPH radical scavenging activity; ferric/ferricyanide (Fe^3+^) reducing antioxidant power (FRAP); total phenolic content (TPC); total flavonoid content (TFC); and dimension, texture, and color values of different quantities of date pit flour (DPF) and Assyrian plum flour (APF) incorporated in flour blends and their gluten-free biscuit samples.

Flour blends					Biscuits										
Samples	DPPH Scavenging Activity IC_50_ (mg/)mL	FRAP IC_50_ (mg/)mL	TPC (mg GAE/100 g DW)	TFC (mg CE/100 g DW)	DPPH Scavenging Activity IC_50_ (mg/)mL	FRAP IC_50_ (mg/)mL	TPC (mg GAE/100 g DW)	TFC (mg CE/100 g DW)	Diameter (mm)	Thickness (mm)	Spread Ratio (mm)	Breaking Force (*n*)	*L**	*a**	*b**
T0	378.21 ± 0.15 ^f^	347.13 ± 4.52 ^f^	38.43 ± 0.11 ^a^	18.67 ± 0.11 ^a^	360.10 ± 3.15 ^d^	340.01 ± 4.52 ^d^	34.73 ± 0.12 ^a^	20.41 ± 0.13 ^a^	41.27 ± 0.56 ^c^	8.17 ± 0.52 ^d^	5.05 ± 0.10 ^d^	21.90 ± 0.39 ^c^	76.25 ± 0.32 ^d^	3.34 ± 0.01 ^a^	25.26 ± 0.10 ^a^
T1	210.40 ± 1.29 ^e^	108.35 ± 0.51 ^e^	75.02 ± 0.18 ^b^	47.30 ± 0.78 ^b^	200.40 ± 1.29 ^c^	102.45 ± 0.4 ^c^	68.74 ± 2.18 ^b^	64.53 ± 1.03 ^b^	41.34 ± 0.40 ^d^	7.76 ± 0.2 ^a^	5.32 ± 0.14 ^e^	19.82 ± 0.2 ^a^	70.14 ± 0.27 ^c^	3.86 ± 0.03 ^b^	25.64 ± 0.12 ^b^
T2	117.62 ± 1.21 ^d^	86.34 ± 0.21 ^d^	102.61 ± 2.31 ^c^	67.94 ± 1.02 ^c^	111.62 ± 1.21 ^b^	78.64 ± 0.3 ^b^	96.21 ± 2.21 ^c^	94.21 ± 1.20 ^c^	41.56 ± 0.88 ^e^	7.06 ± 0.20 ^b^	5.46 ± 0.12 ^c^	20.87 ± 0.23 ^b^	64.21 ± 0.13 ^b^	4.67 ± 0.05 ^c^	26.09 ± 0.17 ^c^
T3	96.04 ± 1.01 ^c^	63.32 ± 0.45 ^c^	132.20 ± 3.42 ^d^	87.27 ± 1.12 ^d^	80.04 ± 1.01 ^a^	58.39 ± 0.15 ^a^	126.49 ± 3.12 ^d^	122.54 ± 1.27 ^d^	40.63 ± 0.51 ^b^	7.78 ± 0.21 ^c^	5.22 ± 0.13 ^b^	23.77 ± 0.30 ^d^	60.34 ± 0.10 ^a^	5.21 ± 0.08 ^d^	26.84 ± 0.19 ^d^
DPF	40.11 ± 0.92 ^b^	26.81 ± 0.13 ^b^	359.37 ± 3.55 ^e^	278.31 ± 1.23 ^e^	-	-	-	-							
APF	32.14 ± 0.42 ^a^	15.62 ± 0.11 ^a^	372.48 ± 3.82 ^f^	292.42 ± 1.34 ^f^	-	-	-	-							

Each value is expressed as mean ± S.D. (*n* = 3); mean followed by different letters in the same column differs significantly (*p* ≤ 0.05). The reported results are based on a dry weight basis (DW), where T0; RF:DPF:APF (100:0:0), T1; RF:DPF:APF (90:5:5), T2; RF:DPF:APF (80:10:10), and T3; RF:DPF:APF (70:15:15). DPPH; 2, 2-diphenyl-1-picrylhydrazyl, *L** value indicates the lightness (100: perfect white /0: perfect black), *a** value indicates (+) redness/(−) greenness, and *b** value indicates (+) yellowness/(−) blueness.

**Table 5 foods-11-02346-t005:** Nutritional and sensory analyses of gluten-free biscuits fortified with different levels of date pit flour (DPF) and Assyrian plum flour (APF).

Nutritional Information								Sensory				
Samples	Moisture Content (%)	Ash (%)	Protein (%)	Fat (%)	Fiber (%)	Carbohydrate (%)	Energy (Kcal/100 g)	Appearance (9*)	Color (9*)	Taste (9*)	Texture (9*)	Overall Acceptability (9*)
T0	4.53 ± 0.12 ^a^	1.63 ± 0.01 ^a^	10.20 ± 0.10 ^a^	22.12 ± 0.20 ^d^	0.69 ± 0.26 ^a^	60.83 ± 0.80 ^d^	483	8.35 ± 0.15 ^c^	7.22 ± 0.17 ^a^	8.35 ± 0.13 ^d^	8.23 ± 0.15 ^b^	8.07 ± 0.17 ^b^
T1	4.67 ± 0.14 ^b^	2.50 ± 0.03 ^b^	12.81 ± 0.12 ^b^	19.41 ± 0.14 ^c^	3.88 ± 0.20 ^b^	56.73 ± 0.42 ^c^	453	8.33 ± 0.12 ^b^	8.43 ± 0.15 ^c^	8.32 ± 0.15 ^b^	8.32 ± 0.13 ^c^	8.36 ± 0.16 ^c^
T2	4.67 ± 0.10 ^b^	3.29 ± 0.01 ^c^	13.11 ± 0.11 ^c^	18.21 ± 0.10 ^b^	7.02 ± 0.11 ^c^	53.07 ± 0.57 ^b^	429	8.36 ± 0.17 ^cd^	8.62 ± 0.15 ^d^	8.34 ± 0.12 ^cd^	8.38 ± 0.17 ^d^	8.42 ± 0.19 ^d^
T3	4.68 ± 0.11 ^bc^	4.11 ± 0.02 ^d^	13.73 ± 0.15 ^d^	17.18 ± 0.21 ^a^	10.20 ± 0.21 ^d^	50.10 ± 0.64 ^a^	410	7.52 ± 0.13 ^a^	7.89 ± 0.14 ^b^	7.25 ± 0.10 ^a^	7.73 ± 0.12 ^a^	7.13 ± 0.14 ^a^

Each value is expressed as mean ± S.D. (*n* = 3); mean followed by different letters in the same column differs significantly (*p* ≤ 0.05), where T0; RF: DPF: APF (100:0:0), T1; RF: DPF: APF (90:5:5), T2; RF: DPF: APF (80:10:10), and T3; RF: DPF: APF (70:15:15). Each value is expressed as mean ± standard deviation. (*n*: 95 semi-trained panelists). Mean followed by different letters in the same column differs significantly (*p* ≤ 0.05). 9* Points hedonic scale: 9; like extremely, 8; like very much, 7; like moderately, 6; like slightly, 5; neither like nor dislike, 4; dislike slightly, 3; dislike moderately, 2; dislike very much, 1; dislike extremely.

## Data Availability

The data presented in this study are available on request from the corresponding author.

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
