# Peer review of "The Impact of Innovative Plant Sources (Cordia myxa L. Fruit (Assyrian Plum) and Phoenix dactylifera L. Biowaste (Date Pit)) on the Physicochemical, Microstructural, Nutritional, and Sensorial Properties of Gluten-Free Biscuits"

_foods, 2022, doi:10.3390/foods11152346_

Round 1
Reviewer 1 Report
The study is very interesting and well conducted. A variety of tests and correlations made it possible to reach a better conclusion for the study.
I just have some comments to improve the manuscript
introduction
line 50 - Where are the references for this sentence?
Lines 97 - 99 - it doesnt belong in the introduction.
lines 80-81 -It goes straight from rice to the fruits. It is lacking a connection. My advice is to show through studies that rice is poor in bioactive compounds and fiber, and it is important to look for alternatives to improve it.
Materials
2.3 - Why did you choose these proportions of DPF and APF?
line 208 - take off "each batch"
Table 1 - The amount of water is different from T0 to T3 - Why?
Results and discussion
Line 321 - In this sense.... due to the higher concentration of lipids in DPF and APF" - shouldn´t be T1, T2 and T3?
lines 326 - 328 - should come first them the composition.
Since APF has more fiber, why din´t you use more of this flour?
Table 5 - improve the Fat column
line 737 - take out - "was"
Besides these comments, the manuscript is well presented.
Author Response
I, as the author highly grateful for the respected and potential reviewer who has devoted time to read and give us his valuable comments and insights regarding our manuscript, which I'm sure has upgraded the quality of our research output, particularly the manuscript submitted to FOODS for possible publication. The following are respected reviewer comments and author responses:
Introduction
1)Line 50 - Where are the references for this sentence?
Answer) Reference inserted Line # 71
2) Lines 97 - 99 - it doesnt belong in the introduction.
Answer) Sentence deleted
3)lines 80-81 -It goes straight from rice to the fruits. It is lacking a connection. My advice is to show through studies that rice is poor in bioactive compounds and fiber, and it is important to look for alternatives to improve it.
Answer) Sentence corrected Line #102 to 105
Materials
4) 2.3 - Why did you choose these proportions of DPF and APF?
Answer) We have done the pretrials for the biscuits formulations up to 40% of addition of DPF and APF and observed that only up to 30%, i.e 15 % each of DPF and APF was desirable for the physical and sensorial properties of biscuits.
5) line 208 - take off "each batch"
Answer) Sentence corrected Line # 236 to 237
6) Table 1 - The amount of water is different from T0 to T3 - Why?
Answer) As the concentration of DPF and APF increased in the dough or biscuits formulation, more amount of water was required for uniform blending, mixing, and hydration of all the ingredients. The increased water absorption / incresesd water requirement was due to the high level of fibers and proteins in DPF and APF.
Results and discussion
7) Line 321 - In this sense.... due to the higher concentration of lipids in DPF and APF" - shouldn´t be T1, T2 and T3?
Answer) Sentence corrected Line # 354
8) lines 326 - 328 - should come first them the composition.
Answer) In this scentence, we have discussed the milling conditions, i.e., wet and dry, and correlated those conditions with the damaged starch content. The composition of biscuits was discussed step by step regarding each component.
9) Since APF has more fiber, why din´t you use more of this flour?
Answer) DPF has more fiber than APF as reported in Table 2, and we have kept equal proportions ( for both DPF and APF) for each of the formulations.
10) Table 5 - improve the Fat column
Answer) Table 5 corrected
11) line 737 - take out - "was"
Answer) Scentence corrected Line # 775

Reviewer 2 Report
Comments for Foods-1856751
The paper reports a method to prepare the gluten-free biscuits with rice flour incorporated with Assyrian plum flour and bio-waste Date pit flour. The contents of protein, crude fiber and antioxidants in those biscuits are higher than that of control group without sacrificing taste. This is interesting and meaningful. Some shortcomings are as following:
1. Pasting, Thermal, micrograph, antioxidant activity and other properties are not described in the abstract.
2. The authors should discuss whether harmful substances would be produced by polyphenols during baking, especially those substances with dark color.
3. In line 703-708, whether amylose-lipid complexes produced have effects on storage of biscuits?
4. Authors should discuss the interaction of components incorporated and its effects on nutrition of biscuits.
5. Line 618-620, there is a grammar mistake in the sentence of “The observed variations in the values of TFC was may be due to the formation of brown-colored pigments……” “was” should be deleted.
6. Line 663-665, there is a grammar mistake in the sentence of “The diameter of biscuit samples prepared from the flour blends of T1 and T2 was higher compared to T0 and T3……” should be “The diameter of biscuit samples prepared from the flour blends of T1 and T2 was higher than those of T0 and T3” should be deleted.
Author Response
I, as the author highly grateful for the respected and potential reviewer who has devoted time to read and give us his valuable comments and insights regarding our manuscript, which I'm sure has upgraded the quality of our research output, particularly the manuscript submitted to FOODS for possible publication. The following are respected reviewer comments and author responses:
- Pasting, Thermal, micrograph, antioxidant activity, and other properties are not described in the abstract.
Answer) Abstract corrected
- The authors should discuss whether harmful substances would be produced by polyphenols during baking, especially those substances with dark color.
Answer) Sentence inserted Line # 137 to 140
- In line 703-708, whether amylose-lipid complexes produced have effects on storage of biscuits?
Answer) Sentence inserted Line # 744 to 746
- Authors should discuss the interaction of components incorporated and its effects on nutrition of biscuits.
Answer) Sentences are inserted Line # 746 to 749
- Line 618-620, there is a grammar mistake in the sentence of “The observed variations in the values of TFC was may be due to the formation of brown-colored pigments……” “was” should be deleted.
Answer) Sentence corrected Line # 654
- Line 663-665, there is a grammar mistake in the sentence of “The diameter of biscuit samples prepared from the flour blends of T1 and T2 was higher compared to T0 and T3……” should be “The diameter of biscuit samples prepared from the flour blends of T1 and T2 was higher than those of T0 and T3” should be deleted.
Answer) Sentence corrected Line # 673 to 675

Round 2
Reviewer 2 Report
Authours have corrected all mistakes, so I recommand acceptence of paper at present status.